# Intestine Health and Barrier Function in Fattening Rabbits Fed Bovine Colostrum

**DOI:** 10.3390/vetsci10110657

**Published:** 2023-11-15

**Authors:** Lucia Aidos, Margherita Pallaoro, Giorgio Mirra, Valentina Serra, Marta Castrica, Stella Agradi, Giulio Curone, Daniele Vigo, Federica Riva, Claudia Maria Balzaretti, Roberta De Bellis, Grazia Pastorelli, Gabriele Brecchia, Silvia Clotilde Modina, Alessia Di Giancamillo

**Affiliations:** 1Department of Veterinary Medicine and Animal Sciences, University of Milan, Via dell’Università 6, 26900 Lodi, Italy; lucia.aidos@unimi.it (L.A.); margherita.pallaoro@unimi.it (M.P.); giorgio.mirra@unimi.it (G.M.); valentina.serra@unimi.it (V.S.); stella.agradi@unimi.it (S.A.); giulio.curone@unimi.it (G.C.); daniele.vigo@unimi.it (D.V.); federica.riva@unimi.it (F.R.); claudia.balzaretti@unimi.it (C.M.B.); grazia.pastorelli@unimi.it (G.P.); gabriele.brecchia@unimi.it (G.B.); silvia.modina@unimi.it (S.C.M.); 2Dipartimento di Biomedicina Comparata e Alimentazione—BCA, University of Padua, Viale dell’Università, 16, 35020 Legnaro, Italy; marta.castrica@unipd.it; 3Department of Biomolecular Sciences, University of Urbino “Carlo Bo”, Via A. Saffi 2, 61029 Urbino, Italy; roberta.debellis@uniurb.it; 4Department of Biomedical Sciences for Health, University of Milan, Via Mangiagalli 31, 20133 Milan, Italy

**Keywords:** intestinal barrier, rabbits, intestinal health, zonulin, enteric nervous system, bovine colostrum

## Abstract

**Simple Summary:**

In rabbit breeding, weaning constitutes a delicate event for the digestive system, and because of the transition from the mother’s milk to a solid diet, nonspecific enteritis and gastrointestinal infections may occur. Enteropathies in rabbits are responsible for high mortality rates but also for decreased growth and the consequent economic losses. It is therefore of major importance to understand the potential influence of the dietary factors on the gastrointestinal functions’ development at weaning, in particular on the integrity of the intestinal barrier, to ensure a reduction in mortality and diseases and an increased growth rate in the subsequent fattening period. The present study investigated the effects of a bovine-colostrum-supplemented diet in the intestine of post-weaning rabbits. Our results indicate that an intermediate level of bovine colostrum diet supplementation may have a positive trophic effect in weaning rabbits.

**Abstract:**

The permeability of the immature intestine is higher in newborns than in adults; a damaged gut barrier in young animals increases the susceptibility to digestive and infectious diseases later in life. It is therefore of major importance to avoid impairment of the intestinal barrier, specifically in a delicate phase of development, such as weaning. This study aimed to evaluate the effects of bovine colostrum supplementation on the intestinal barrier, such as the intestinal morphology and proliferation level and tight junctions expression (zonulin) and enteric nervous system (ENS) inflammation status (through the expression of PGP9.5 and GFAP) in fattening rabbits. Rabbits of 35 days of age were randomly divided into three groups (n = 13) based on the dietary administration: commercial feed (control group, CTR) and commercial feed supplemented with 2.5% and 5% bovine colostrum (BC1 and BC2 groups, respectively). Rabbits receiving the BC1 diet showed a tendency to have better duodenum morphology and higher proliferation rates (*p* < 0.001) than the control group. An evaluation of the zonulin expression showed that it was higher in the BC2 group, suggesting increased permeability, which was partially confirmed by the expression of GFAP. Our results suggest that adding 2.5% BC into the diet could be a good compromise between intestinal morphology and permeability, since rabbits fed the highest inclusion level of BC showed signs of higher intestinal permeability.

## 1. Introduction

Weaning represents a critical period in rabbits, as in other species: rabbit kits are separated from their mothers and maternal milk is replaced by solid feed [1,2]. These environmental and physiological changes constitute stressful events that can lead to nonspecific enteritis and gastrointestinal infections [3]. Enteropathies are responsible for high rabbit breeding mortality rates, and decreased growth, leading to consequent economic losses [4].

The maintenance of intestinal health is a fundamental and multifactorial process that includes (i) endogenous factors that depend on the intestinal mucosa, with its absorptive and barrier functions, and (ii) exogenous factors such as the husbandry system. The intestinal epithelium changes gradually from birth to maturity and is mainly made of epithelial cells, called enterocytes, which exert their absorptive and barrier functions. 

The correct growth and development of the intestinal anatomy is fundamental not only for the absorption of nutrients but also for the function of the barrier that acts against pathogenic microorganisms [5]. The intestinal barrier is composed of a multilayer system, which, in addition to the already mentioned intestinal epithelial cells, is made up of the immune and the enteric nervous system (ENS). One of the most critical mechanisms is the establishment of a permeable barrier, which is regulated predominantly by tight junctions (TJs), which consist of numerous intracellular and apical intercellular membrane proteins. Damage to the intestinal barrier’s integrity may result in the passage of pathogens and harmful substances, leading to systemic inflammation, a condition known as “leaky gut” or intestinal wall leakage syndrome [6]. Intestinal zonulin is a physiological intestinal permeability modulator and constitutes a valid marker of intestinal permeability [7]. It is now known that weaning stress can cause intestinal barrier dysfunction by inducing inflammation, as recently reviewed by Bivolarski and Vachkova [4].

Furthermore, in health and disease conditions, the ENS affects epithelial barrier function, even if the influence of weaning on ENS function has not been extensively studied in rabbits. The ENS is composed of millions of neurons and glial cells that are arranged in interconnected ganglia embedded within the intestinal wall. It can be divided into two distinct ganglionated neuronal plexuses: the myenteric or Auerbach’s (located between the longitudinal and the circular muscle layers), and the submucosal or Meissner’s (located in the submucosa between the smooth muscle layers and mucosa). The myenteric plexus coordinates the movement of the muscle underlying the propulsion of content, whereas the submucosal plexus is involved in secretion and absorption [8]. Within the ganglia, the glial cells interdigitate with neurons and are morphologically equivalent to the astrocytes and microglia in the central nervous system. The development of the ENS does not end with birth. Indeed, terminal differentiation of enteric neurons proceeds after birth, in which neurons respond to perturbations of any severity [9,10]. It is therefore to be expected a certain degree of plasticity of the ENS and a relevant influence of the internal and external conditions [11]. Recently, it was found that the role of the enteric glia is more than just the nutritive support for enteric neurons, as it plays a major role in the regulation of inflammatory events in the intestine [12]. An investigation of ENS morphology may indicate the intestine’s inflammatory condition.

Over the last few years, an increasing number of clinical trials have achieved significant results indicating bovine colostrum (BC) as a promising nutraceutical that can prevent or alleviate various diseases in humans and animals as well [13,14,15,16,17,18,19,20,21,22]. BC is the first biological fluid produced by the mammary gland of cows after calving in the first 4 days of lactation, before becoming mature milk [23]. If compared with mature milk, BC presents higher levels of protein, fat, vitamins, and minerals [13], as well as growth factors, including insulin, insulin-like growth factor-I (IGF-I), transforming growth factor beta (TGF-β), and epidermal growth factor (EGF) [24,25]. Some of these growth factors show mitogenic or trophic potential, and, therefore, it has been hypothesized that their ingestion through the colostrum may improve tissue growth and development of the newborn [26]. Studies show that EGF has an important role in intestinal barrier integritys by regulating the components of tight junctions [27,28]. TGF-β plays a major role in intestinal immunity by suppressing inflammatory responses to luminal bacterial antigens and by contributing to the induction of immune tolerance [29,30]. An additional benefit of colostrum is the high content of immunoglobulins and antimicrobial factors that have an important role in preventing intestinal inflammation [25], although this topic is not part of the present study.

In the zootechnical field, the use of BC has shown positive effects on immune status [31,32], on the reduction of the incidence of some pathologies and mortality [22,33], and on the improvement of the productive performance of various livestock animals, such as pigs, broilers, lambs, and calves [19,34]. Intestinal health assessments in livestock regard mostly piglets [35,36,37] but also lambs [19] and calves [16]. Overall, there is strong evidence that supplementation with BC has a positive effect on preventing intestinal inflammation. Still, there is limited research on livestock animals concerning the effect of BC on intestinal health, while in humans there are many studies that demonstrate a beneficial effect of BC supplementation on the intestine in terms of permeability and inflammation [13,25,38,39,40,41,42,43,44,45]. 

This study aimed to determine the effect of dietary supplementation with two different concentrations of BC, 2.5% and 5%, on the intestinal health status of weaned rabbits. The barrier function status was investigated through an evaluation of the intestinal morphology and proliferation level, zonulin expression, and ENS inflammation status. Rabbits were selected for this study as they constitute an interesting species bred for research, pets, and livestock [46,47].

## 2. Materials and Methods

### 2.1. Animals and Diets

The experiment was carried out at the Department of Agricultural, Food, and Environmental Science of the University of Perugia, following Legislative Decree No. 146, implementing the Directive 98/58/EC of 20 July 1998 concerning the protection of animals kept for farming purposes and received the approval of the Ethics Committee of the University of Milan, with the approval Code: OPBA_42_2021, 7 May 2021.

Thirty-nine young rabbits were divided into three groups (n = 13) according to the diet administered after weaning (35 days of age): control group (CTR), fed with a commercial diet; BC1 and BC2 groups, fed the diet supplemented with 2.5% and 5% BC, respectively. The choice of these BC groups was based on the dosage used in two previously published works on rabbits by our research group [14,48] but also on works on other animal species, including laboratory animals [15,49] and humans [50,51]. The relationships were determined using the weight of the rabbits and the feed consumption, which led to the choice of two levels of integration: 2.5%, the standard dose, and 5%, the higher dose. The environmental conditions in the rabbitry were 16 h of light and 8 h of dark per day (maximal intensity being 80 lx), air temperature between 20 and 24 °C, and 65% humidity. The whole experiment lasted 91 days. During the trial, the animals were placed in individual cages that were equipped with a feeder and automatic watering system. The feed and the water were administered ad libitum. The diets included no antibiotics, anticoccidials, or other medications, and the supplement was included in the basal, pelleted diet. The two dosages of the bovine colostrum were chosen according to Serra et al. [18], Castrica et al. [14], All analyses of the experimental diets were performed in accordance with the methods of the Association of Analytical Chemists (AOAC). The animals were weighed weekly from weaning to slaughter, and the carcass weight was also registered.

### 2.2. Intestine Histology and Histometry for Trophic Activity

Portions of the intestine (duodenum, jejunum, ileum, cecum, and proximal colon) were collected immediately after the euthanasia (by electrocution) of each animal. All samples were fixed in 4% paraformaldehyde in 0.01 mol/L phosphate-buffered saline (PBS) at pH 7.4 for 24 h at 4 °C, dehydrated in a graded series of ethanol, cleared with xylene, and embedded in paraffin. Microtome sections (4 μm thick) of the intestine portions were stained with hematoxylin–eosin (HE) to establish structural details.

On the HE-stained sections of the duodenum, jejunum, and ileum, the height of the intestinal villi (V) (10 villi measured per section) and the depth of intestinal crypts (C) (10 crypts measured per section) were measured and calculated using image analysis software (Proview, version 3.7, Optika, Ponteranica, Italy). The ratio of villi/crypts (V/C) was also calculated. On the HE-stained sections of the cecum and colon, the depth of the crypts (10 crypts were measured per section) was measured and calculated using image analysis software (Proview, Optika, Italy).

### 2.3. Intestine Histochemistry: AB-PAS

Other sections of the intestine were stained to determine the mucin profile. For this purpose, the following histochemistry stain was applied: Alcian blue 8GX pH 2.5-periodic acid Schiff (AB-PAS) sequence, revealing neutral (PAS-reactive, purple-stained) and acid (AB-reactive, azure-stained) glycoconjugates, according to Rossi et al. [52]. On all portions of the intestine, and for each section, the mucus thickness was measured in ten randomly chosen villi per animal and calculated at 100× using image analysis software (Proview, Optika, Italy).

### 2.4. Immunohistochemistry (PCNA for Trophic Activity)

Immunostaining on other duodenum sections was performed to detect proliferating cell nuclear antigen (PCNA). This was performed only on the duodenum, because it was the only portion of the intestine in which morphometric differences were found. Briefly, endogenous peroxidase activity was blocked by incubating the sections in 3% H_2_O_2_ in PBS. Non-specific binding sites were blocked by incubating the sections in normal mouse serum (Dakocytomation, Milan, Italy). Mouse monoclonal anti-PCNA (dilution 1:100, clone PC10, Sigma-Aldrich, Milan, Italy) antibodies were applied overnight at room temperature. The primary antisera were diluted with a 0.05 MTris–HCl-buffered saline pH 7.4 (TBS: 0.05 M, pH 7.4, 0.55 M NaCl). After treatment with the primary antibody, the antigen–antibody complexes were detected with a peroxidase-conjugated polymer that carries secondary antibody molecules directed against mouse immunoglobulins for PCNA (EnVisionTM+, DakoCytomationDenmark A/S, Glostrup, Denmark), applied for 2 h at room temperature. Peroxidase activity was then detected with diaminobenzidine (DAB, DakoCytomationDenmark A/S) as the substrate. All sections were weakly counterstained with Mayer’s hematoxylin, dehydrated, and permanently mounted. The specificity tests for the used antibodies were performed by incubating other sections in parallel with (i) TBS instead of the specific primary antibodies and (ii) TBS instead of the secondary antibodies. The results of these controls were always negative (i.e., staining was abolished). Photomicrographs were taken with an Olympus BX51 microscope (Evident Corporation, Tokyo, Japan), equipped with a digital camera, and the final magnifications were calculated.

The relative number of PCNA-immunopositive cells was evaluated by counting the immunopositive nuclei and total nuclei, to obtain the number of proliferating cells over the total number of cells, in 10 randomly selected crypts per animal.

### 2.5. Immunofluorescence

#### 2.5.1. Single Immunofluorescence: Zonulin-1 for Intestinal Permeability

Immunofluorescence was performed on other duodenum sections, to assess the intestinal permeability, by quantifying zonulin (ZO-1), which is a modulator of tight junctions. After rehydration, for the antigen retrieval, the duodenum sections were microwaved in citrate buffer at pH 6 for 5 min at 600 W. After cooling, the sections were washed three times in phosphate-buffered saline (PBS, pH 7.4) and treated with the Avidin–Biotin blocking kit solution (Vector Laboratories Inc., Burlingame, CA, USA). Sections were then incubated with the primary antiserum, 1:100 ZO-1 antibody (Cat. No. 33-9100, ZO1-1A212 monoclonal antibody, Invitrogen, MA, USA) for 24 h at room temperature. Afterwards, the sections were washed in PBS and incubated with 10 µg/mL goat biotinylated anti-mouse IgG (Vector Laboratories Inc., Newark, NJ, USA) for 2 h at room temperature. After washing twice in PBS, the sections were treated with Fluorescein–Avidin D (Vector Laboratories Inc., Newark, NJ, USA), 10 µg/mL in NaHCO_3_, 0.1 M, pH 8.5, 0.15 M NaCl for 2 h at room temperature. Lastly, the sections were fixed in Vectashield Mounting Medium with DAPI (SKU H-1200-10, Vector Laboratories Inc., Newark, NJ, USA) and examined using a confocal laser scanning microscope (FluoView FV300; Olympus, Evident Corporation, Tokyo, Japan). Argon/helio–neon–green lasers with excitation and barrier filters set for rhodamine were used for exciting the immuno-fluo-reactive structures. Images involving overlapping fluorescence were obtained by sequentially acquiring the image cut of each laser excitation. The omission of the primary antibody during the first incubation step guaranteed the absence of cross-reactivity with the secondary antibody.

For the quantification of each of the ZO-1 immunofluorescences, the duodenum sections were examined using a confocal laser microscope (Olympus FV300-IX, Evident Corporation, Tokyo, Japan, equipped with argon:helium:neon lasers). For the image analyses, the FluoView software (FV10-ASW Version 4.2b, Olympus) was used. Excitation and barrier filters were set for rhodamine. In order to compare fluorescence intensities of various samples, the laser power and photomultiplier tube voltage were constant. Images were digitized under a constant gain and laser offset, and no postcapture alterations were performed. Before quantification, the images were digitally zoomed 2.0 times according to [53]. Five section areas of the epithelium that contained the largest and brightest immunofluorescence for each sample were selected for measurement. The areas to be assessed were defined manually and used to normalize each peak intensity. The calculated mean fluorescence intensity was obtained for each of the selected section areas according to [53]. The pixel intensity was determined using the histogram/area functions of the FluoView software (FV10-ASW Version 4.2b, Olympus, which assigned the gray levels (GLs) within a 0–256 Gy scale. The data are presented as the mean fluorescence intensity.

#### 2.5.2. Double Immunofluorescence: PGP9.5 and GFAP for Inflammation and Neuronal Plasticity

On other duodenum sections, double immunofluorescence was performed to characterize the enteric nervous system ganglia, both myenteric and submucosal according to our previous study [54]. After rehydrating the tissue sections, antigen retrieval was performed with heat-induced microwaving in citrate buffer at pH 6 for 5 min with the microwaves at 500 W, followed by cooling twice. The sections were washed three times in phosphate-buffered saline (PBS, pH 7.4) and treated with the Avidin–Biotin blocking kit solution (Vector Laboratories Inc., Burlingame, CA, USA). Afterwards, the sections were incubated with the first-step primary antiserum 1:100 anti-PGP9.5 (Anti-Protein Gene Product 9.5 monoclonal antibody 31A3, Cat. No. MA1-83428 Invitrogen, MA, USA) for 24 h at room temperature. After washing in PBS, the sections were incubated with 10 µg/mL of a solution of goat biotinylated anti-mouse IgG (Vector Laboratories Inc.) in PBS for 2 h at room temperature. Then, the sections were treated with 10 µg/mL Fluorescein–Avidin D (Vector Laboratories Inc., Newark, NJ, USA) in 0.1 M NaHCO3 at pH 8.5 and 0.15 M NaCl for 2 h at room temperature after rinsing with PBS. The second step of the double immunofluorescence procedure consisted of treating the sections with 1:100 anti-GFAP (Glial fibrillary acidic protein monoclonal antibody aA5, ab4648, Abcam, Cambridge, UK) overnight. The sections were washed in PBS for 10 min and incubated with 10 µg/mL goat biotinylated anti-mouse IgG (Vector Laboratories Inc., Newark, NJ, USA) for 2 h at room temperature. Afterwards, the sections were rinsed twice in PBS and treated with 10 µg/mL Rhodamine–Avidin D (Vector Laboratories Inc. Newark, NJ, USA) in 0.1 M NaHCO3 at pH 8.5 with 0.15 M NaCl for 2 h at 18–20 °C. Lastly, tissue sections were fixed in Vectashield Mounting Medium with DAPI (SKU H-1200-10, Vector Laboratories Inc., Newark, NJ, USA) and examined using a confocal laser scanning microscope (FluoView FV300; Olympus). Argon/helio–neon–green lasers with excitation and barrier filters set for rhodamine were used for exciting the immuno-fluo-reactive structures. Images involving overlapping fluorescence were obtained by sequentially acquiring the image cut of each laser excitation. The omission of the primary antibody during the first incubation step guaranteed the absence of cross-reactivity with the secondary antibody.

For the quantification of each of the two antibodies’ immunofluorescence, the duodenum sections were examined similarly as mentioned above for ZO-1. Briefly, excitation and barrier filters were set for fluorescein and rhodamine. The laser power and photomultiplier tube voltage were constant so that the fluorescence intensities of various samples could be compared. Images were digitized under constant gain and laser offset, with no post capture modifications. Before quantification, the images were digitally zoomed 1.0 times according to [53]. The five myenteric and submucosal plexuses that contained the largest and brightest immunofluorescence for each sample were selected for measurement. The plexus areas to be assessed were defined manually and used to normalize each peak intensity. The calculated mean fluorescence intensity was obtained for each of the selected plexus areas according to [53]. The pixel intensity was determined using the histogram/area functions of the FluoView software, which assigned the gray levels (GL) within a 0–256 Gy scale. Data are presented as mean fluorescence intensity.

The myenteric and submucosal plexuses were counted and normalized by the intestinal area, which was calculated using the FluoView software (FV10-ASW Version 4.2b, Olympus). 

### 2.6. Statistical Analyses 

One-way ANOVA was performed when the data were normally distributed; otherwise, the Kruskal–Wallis test was applied. To evaluate the body weight of rabbits, the model included the group, time, and their interaction. Moreover, the body weight (BW) at weaning and the sex for BW after weaning were included as covariates. The b-parameter with its standard error was reported for parameters included as covariates (time). The body weight (BW) of the rabbits was evaluated using a linear mixed model including time as a repeated effect and rabbits as subjects. The model evaluated the effect of the group (3 levels), time (5 levels), and their interaction. Moreover, the BW at weaning and the sex of each rabbit were included as covariates. The b-parameter with its standard error was reported for parameters included as continuous variables (i.e., BW at weaning). The data were analyzed with SPSS Statistics version 25 (IBM, SPSS Inc., Chicago, IL, USA). The experimental unit was the single rabbit. For all analyses performed, except for the body weight, the value for each animal was the mean of all measures obtained. The data are presented as the means ± S.E.M. Differences between the means were considered significant at *p* < 0.05.

## 3. Results

### 3.1. Zootechnical Parameters—Rabbits Growth

Rabbits from the CTR group showed a higher BW than animals from the supplemented groups (Figure 1). As previously reported, the statistical model was corrected for the weight at weaning and sex of the rabbits, showing that the BC2 group presented the greatest weight gain (estimated marginal means of BW during growth: 1.31 ± 0.1, 1.29 ± 0.1, and 1.34 ± 0.1 kg for CTR, BC1, and BC2, respectively; *p* < 0.05). Apparently, these results seem to be incoherent, but this is due to the remarkable effect of the weight at weaning (b = 0.83 ± 0.04; *p* < 0.001) and the influence of the sex (*p* = 0.047) of the rabbits. Moreover, the univariate test for each day showed that after day 56, the differences between the groups were not significant (Figure 1).

### 3.2. Small and Large Intestine Histology and Histometry

Histological evaluation of the whole tracts of the intestine revealed that dietary supplementation with BC did not cause detrimental effects on the intestine structure in all experimental groups. We found no statistical differences regarding the morphometric parameters analyzed in any of the intestinal tracts. However, at the duodenum level, the BC1 group showed a tendency to have a higher villi height in respect to the CTR (Figure 2A, BC1 > CTR, *p* = 0.07).

### 3.3. Small and Large Intestine Histochemistry: AB-PAS

The histochemical analyses show that the duodenum, jejunum, and ileum goblet cells contained mixtures of mixed (purple) and acidic (blue) glycoconjugates in the villi (Figure 3A–C, arrows) in either the control or treated rabbits, regardless of the BC inclusion level. No differences were found between the experimental groups in the cecum and the colon as well; in these tracts, the goblet cells contained neutral (magenta) and acidic (blue) glycoconjugates (Figure 3D,E, arrows). 

In all portions of the intestine, mucus was made of mixed glycoconjugates (Figure 4A), and there were no significant differences in the mucus thickness (Figure 4B–F).

### 3.4. Duodenum Immunohistochemistry (PCNA for Trophic Activity)

Immunostained nuclei in the duodenum were mostly detected in epithelial cells of intestinal crypts and a few in epithelial cells of the villi (Figure 5A, arrows). 

The quantification of the PCNA-immunostained nuclei showed that rabbits fed diets with BC presented more cell proliferation than the animals fed the CTR diet at the duodenum level (Figure 5B; BC1 > CTR, *p* < 0.001; BC2 > CTR, *p* < 0.01). 

### 3.5. Immunofluorescence

#### 3.5.1. Single Immunofluorescence: Zonulin-1 for Duodenum Permeability 

ZO-1 was located at the apical end of the enterocytes, often as dots on the cell surface, on the top of the villi (Figure 6A, white arrows). From the CTR group to the BC2, there was a growing increase in ZO-1 expression in the duodenum, but it was statistically different only between the CTR group and BC2 (Figure 6B, *p* < 0.05).

#### 3.5.2. Duodenum Double Immunofluorescence: PGP9.5 and GFAP for Inflammation and Neuronal Plasticity 

Neuronal plexuses of the duodenum were observed both in the submucosa (Figure 7A, asterisk) and between the circular and longitudinal muscle layers (Figure 7B, asterisk). Ganglia of the submucosal and myenteric plexuses contained a certain number of enteric neurons, which could be observed through the expression of PGP9.5 (Figure 7A,B, green color); GFAP positivity was observed in small cell bodies and in their cytoplasmic processes (Figure 7A,B, red color).

The number of ganglia in each plexus was not statistically different among treatments (Figure 8).

Considering the intensity of each marker, we found differences in neither the submucosal nor myenteric plexuses for PGP9.5 (Figure 7C,D), but regarding GFAP, we found that the expression in the submucosal plexuses was significantly higher for the BC2 group compared to both the CTR and BC1 groups (Figure 7E, *p* < 0.05). For what concerns the expression of GFAP in the myenteric plexuses we found a tendency to be higher in the BC2 group respect to the BC1 (Figure 7F, *p* = 0.06).

## 4. Discussion

The intestinal epithelium acts as a physical barrier by limiting bacterial invasion through high proliferation rates, mucus secretion, tight junction formation, and innate immune responses [55]. A damaged intestinal barrier may result in an inflammatory reaction and gastrointestinal diseases [56]. In mammals, the permeability of the immature intestine is higher in newborns than in adults [57]. The post-natal maturation of the intestinal epithelium has long-term consequences for gut homeostasis, and a damaged gut barrier in young animals increases the susceptibility to digestive and infectious diseases later in life [56]. The dietary transition that occurs at weaning is associated with major developmental changes in the intestine, among which is the gut barrier’s formation [58]. Therefore, it is of major importance to develop strategies for weaning animals that support long-term intestinal homeostasis. In our study, the impact of the supplementation of BC on the intestinal barrier in post-weaning rabbits was evaluated, starting from the interesting result that growth was greater in the rabbits in the BC-fed group with the highest inclusion level (BC2). Currently, other than the studies performed by our research group, there is only one another study that evaluated the effect of BC on rabbits intestine [17] that focused, however, on the immune status of the animals. Therefore, we must necessarily compare our results with other animal species, such as pigs. Our results are congruent with previous studies conducted on weaning piglets, where the growth in animals fed with BC supplements was greater [59]. The same result was observed in piglets that received colostrum before weaning as a milk substitute [22,60], therefore suggesting a positive effect on the health status of the animals. Starting from this observation, we now focus on the barrier function of the intestine.

Rabbits fed colostrum showed normal morphological development of the micro-anatomical structure of all intestinal tracts analyzed. The duodenum revealed a direct effect of colostrum, showing higher proliferative activity of cells present in the upper crypts in the treated groups compared to the control. The proliferating activity of the crypts is quite informative: crypts’ bottom base allocate stem cells that generate enterocytes and goblet cells, carrying out cellular renewal and determining the length of the absorptive villi [61]. The greater the proliferating activity, the longer the villi. Indeed, in our study, BC1 showed higher proliferation and a tendency to have longer villi than the CTR. It would seem that the colostrum acts with a trophic effect that is limited to the duodenum, perhaps because it is digested and absorbed directly in this location [62], as observed as well in piglets fed colostrum at weaning [63]. A higher proliferative activity in both experimental groups, BC1 and BC2, may be due to the fact that BC contain several growth factors that show mitogenic or trophic potential [24,25,64,65]. An improvement in gut morphology in terms of villi height was also observed in newborn piglets [36,37] and lambs [19] fed a BC supplement. However, in another study with piglets, animals fed BC showed neither significantly greater crypt depth nor increased cell proliferation [66]. These differences may be due to the level of BC inclusion in the diet used in these studies. Furthermore, we also observed that the mucus produced did not differ in terms of quality and thickness throughout the intestinal tract. We can state with certainty that this first line of defense in contrast to microorganisms infiltration is not damaged regardless of the rabbits’ diet in this study.

Since the morphology of the intestine suggests that colostrum may have a moderate/mild trophic effect, we evaluated the permeability of the intestine. Permeability is regulated by some membrane junction proteins, such as zonulin, which are present in intestinal epithelial cells [12,67], and it has been shown that EGF, which is present in BC, has an important role in intestinal barrier integrity by regulating the components of the tight junctions [27,28]. The results we obtained should be taken seriously given that rabbits receiving the highest level of BC inclusion showed higher ZO-1 expression in the duodenum compared to control animals. Zonulin is an endogenous mediator in the physiological regulation of intercellular tight junctions [7,45,68,69]. Indeed, ZO-1 can reversibly modulate intestinal permeability by regulating intestinal tight junctions [7]. Furthermore, it has been seen that in human intestinal diseases, such as irritable bowel syndrome, nonceliac gluten sensitivity, environmental enteropathy, and necrotizing enterocolitis, there was an increase in the expression of zonulin with a consequent increase in permeability [70]. For this reason, it is possible that the greater expression of zonulin in the BC2 group may indicate an alteration in intestinal permeability.

Another system that plays a key role in controlling intestinal barrier functions is the ENS [71]. Indeed, there is evidence that enteric neurons and enteric glial cells regulate the paracellular permeability of the intestinal barrier, which precedes inflammation [67,71,72,73]. This effect on permeability by the ENS is due to a regulation of ZO-1 expression in intestinal epithelial cells [12,67]. In our study, we found no differences in the expression of a neuronal marker (PGP9.5), but regarding the expression of a glial cell marker (GFAP), we found different expressions between experimental groups. The BC2 group had a significantly higher expression of GFAP marker in the submucosal plexus, as well as a tendency to be greater in the myenteric ones, in comparison to the other two groups. These results for the GFAP marker reveal that the enteric glia activation can be linked to the data observed for the expression of ZO-1. These data may suggest a possible inflammatory activation of the intestine in the group treated with 5% colostrum. Indeed, an increase in GFAP expression was observed following exposure to lipopolysaccharide in rats [74] or after copper exposure in weaning piglets [75], thus confirming the role of the ENS in the response to inflammation. These findings were confirmed by other studies, as reviewed by Grundmann et al. [76]. Enteric glial cells have also been shown to have the ability to strongly inhibit the proliferation of intestinal epithelial cells [77]. In the present study, we did not observe this antiproliferative effect of glial cells on the intestinal epithelium. In a recent study of our group, we found that, although microbial diversity was not strongly modified, bovine colostrum supplementation at 2.5% changed the phylogenetic microbial composition; the families most affected were Clostridia UCG-014, *Barnesiellaceae*, and *Eggerthellaceae* [48]. Vicentini et al. [78] recently demonstrated that the microbiota is essential for the maintenance of ENS integrity by regulating enteric neuronal survival and promoting neurogenesis. This is the reason why we suggest that 2.5% administration may positively modulate the intestinal barrier function: it is a reasonable balance according to all of the collected data, even if these changes should be further investigated.

## 5. Conclusions

Considering all our results, it seems safe to assume that rabbits fed the BC2 diet showed signs of intestinal permeability, greater than the BC1 and CTR groups. Perhaps it would be necessary to evaluate long-term BC2 supplementation or to perform an inflammatory test to clarify the effects of adding BC2 to the rabbits’ diet. A reasonable choice could be to supplement the diet of rabbits after weaning with an intermediate level of BC, for example, 2.5%, which did not result in signs of altered intestinal permeability and, at the same time, demonstrated a trophic effect on the lining of the epithelium of the intestine.

## Figures and Tables

**Figure 1 vetsci-10-00657-f001:**
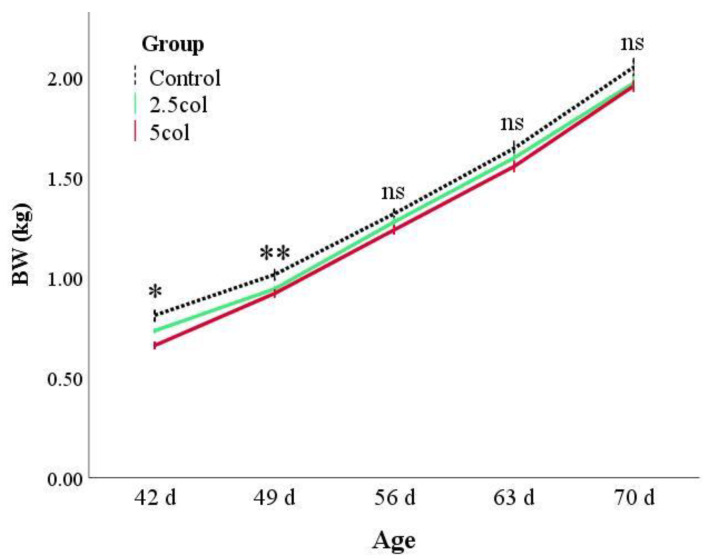
Body weight (BW) expressed in kilograms (kg) in rabbits aged 42, 49, 56, 63, and 70. Values are expressed as the mean ± S.E.M. * *p* < 0.05; ** *p* < 0.01. Group: *p* < 0.001; time: *p* < 0.001; group × time: *p* = 0.088; BW at weaning: *p* < 0.001; sex: *p* = 0.047.

**Figure 2 vetsci-10-00657-f002:**
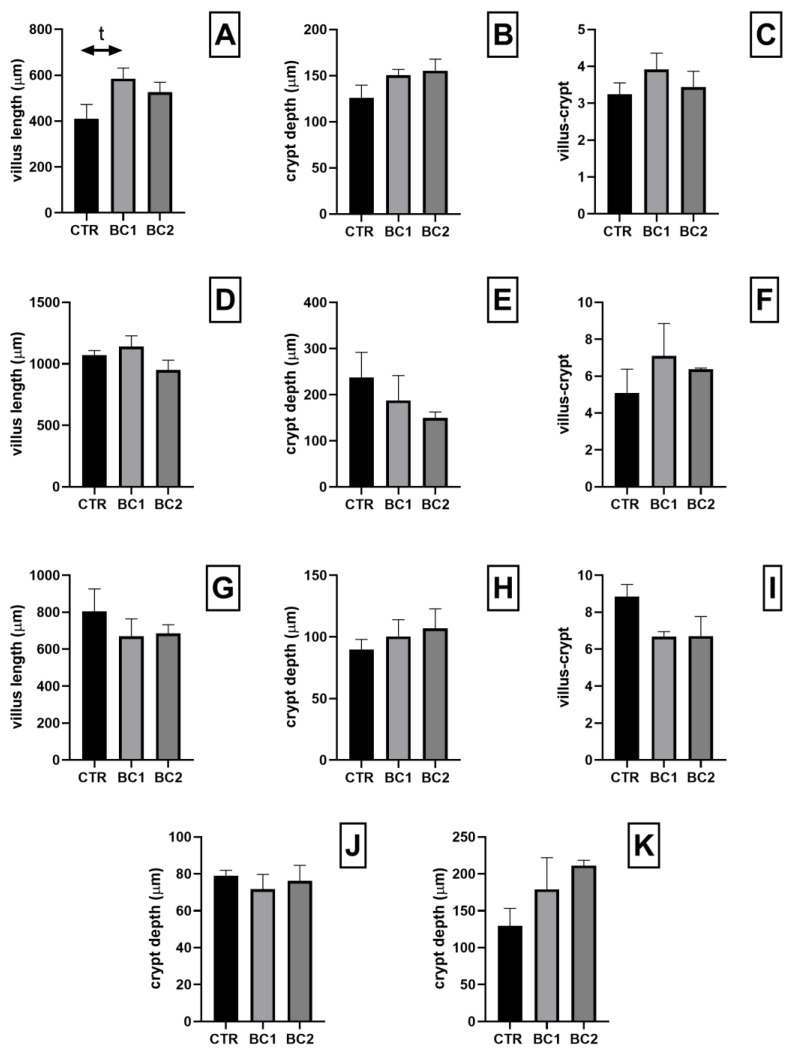
Histometrical analyses: (**A**–**C**) duodenum; (**D**–**F**) jejunum; (**G**–**I**) ileum; (**J**) cecum; (**K**) colon. Villus height, expressed in μm: (**A**) duodenum; (**D**) jejunum; (**G**) ileum. Crypts depth, expressed in μm: (**B**) crypt depth of the duodenum; (**E**) jejunum; (**H**) ileum; (**J**) cecum; (**K**) colon. villus-crypt ratio: (**C**) duodenum; (**F**) jejunum; (**I**) ileum. One-way ANOVA was performed. Values are expressed as the mean ± S.E.M; t, tendency (*p* = 0.07).

**Figure 3 vetsci-10-00657-f003:**
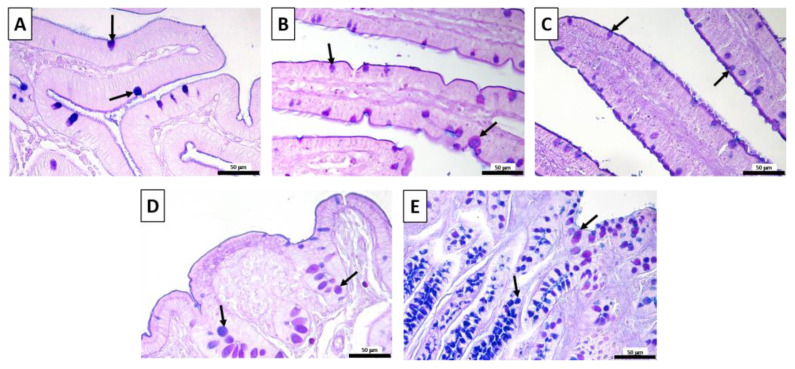
Representative images of the AB-PAS-stained (**A**) duodenum; (**B**) jejunum; (**C**) ileum; (**D**) cecum; (**E**) colon. Goblet cells, arrows; acidic, blue; neutral, magenta; mixed, purple. Scale bars: 50 μm.

**Figure 4 vetsci-10-00657-f004:**
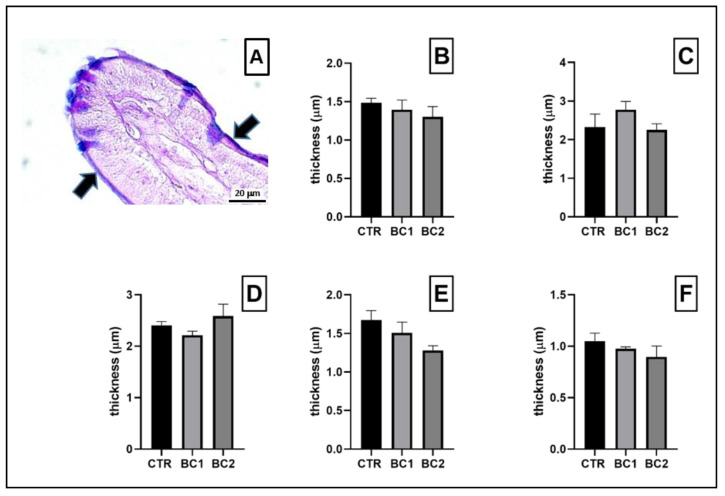
(**A**) Representative image of an enlargement of the tip of a villus in which the mucus layer is visible (bold, black arrows). Scale bar: 20 μm. Quantitative representation of the thickness of the mucus, expressed in μm: (**B**) duodenum; (**C**) jejunum; (**D**) ileum; (**E**) cecum; (**F**) colon. One-way ANOVA was performed. Values are expressed as the mean ± S.E.M.

**Figure 5 vetsci-10-00657-f005:**
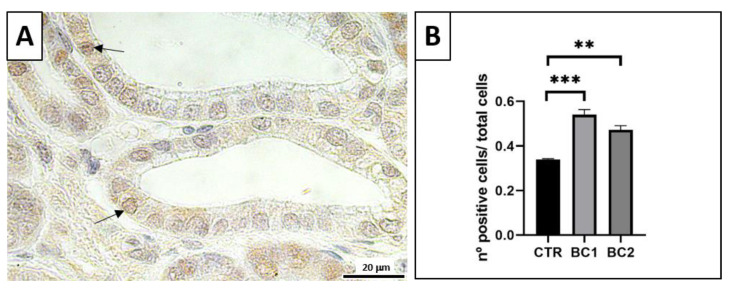
(**A**) Representative image of the PCNA-immunostained duodenum crypts. Positive nuclei, arrows. Scale bar: 20 µm. (**B**) Quantitative representation of PCNA counts in the duodenum, expressed as the number of immunopositive cells over the total number of cells per crypt. One-way ANOVA was performed. Values are expressed as the mean ± S.E.M. ** *p* < 0.01; *** *p* < 0.001.

**Figure 6 vetsci-10-00657-f006:**
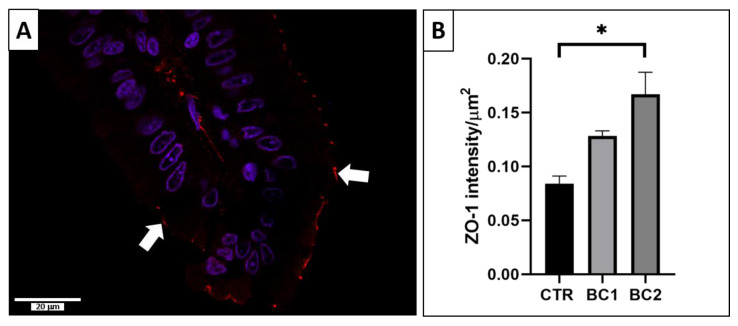
(**A**) Representative image of a ZO-1 immunofluorescence-stained duodenum villus (white arrows). Scale bar: 20 µm. (**B**) Quantitative representation of the expression of ZO-1 in the duodenum, expressed in intensity per µm^2^. One-way ANOVA was performed. Values are expressed as the mean ± S.E.M. * *p* < 0.05.

**Figure 7 vetsci-10-00657-f007:**
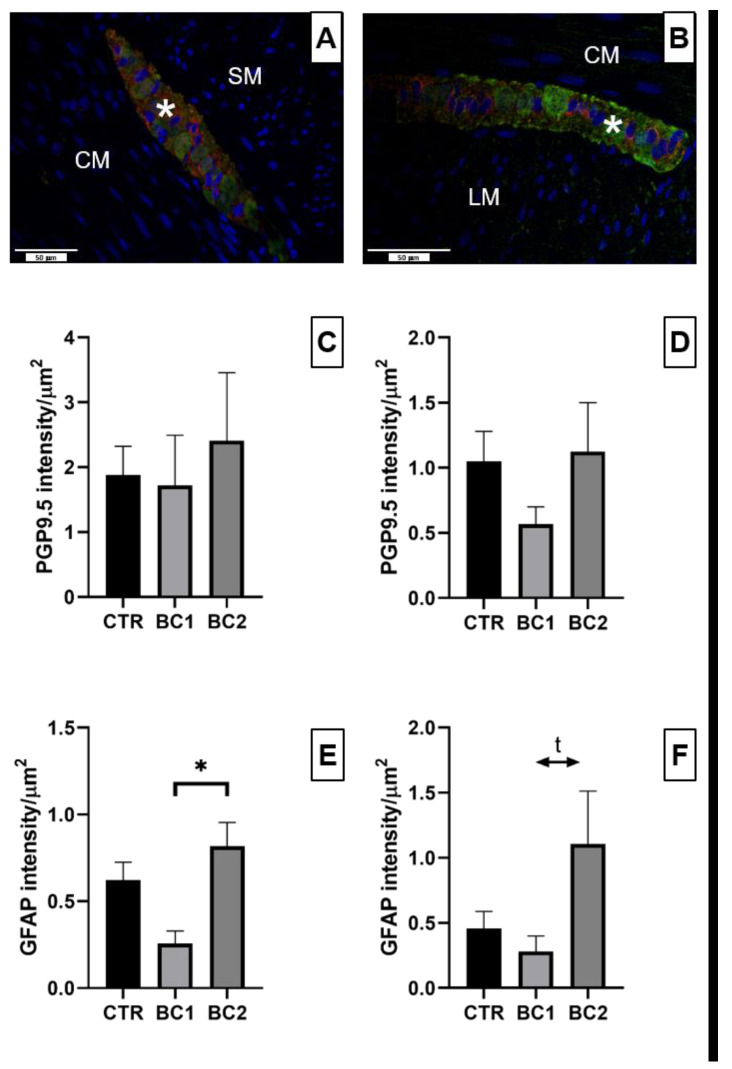
Representative image of (**A**) submucosal and (**B**) myenteric neuronal plexuses of the duodenum stained with PGP9.5 (green color)—GFAP (red color) double immunofluorescence. Ganglions, asterisk; longitudinal muscular layer, LM; circular muscular layer, CM; submucosa, SM. Scale bar: 50 µm. Quantitative representation of the expression of PGP9.5 in the (**C**) submucosal and (**D**) myenteric neuronal ganglia and GFAP in the (**E**) submucosal and (**F**) myenteric neuronal ganglia in the duodenum, expressed in intensity per µm^2^. One-way ANOVA was performed. Values are expressed as the mean ± S.E.M. * = *p* < 0.05; t, tendency (*p* = 0.06).

**Figure 8 vetsci-10-00657-f008:**
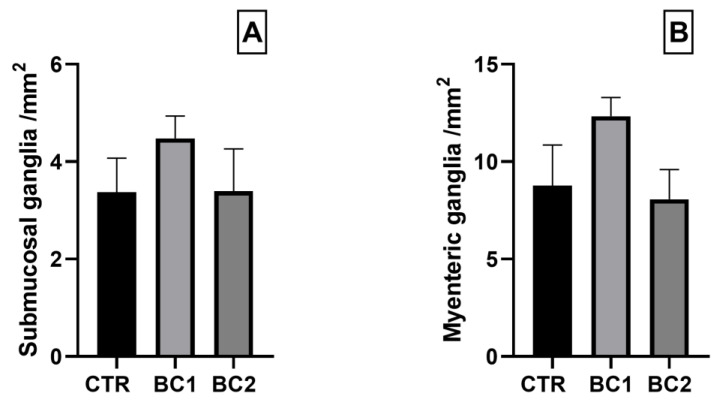
Number of ganglia per section area of the (**A**) submucosal and (**B**) myenteric plexuses of the duodenum. One-way ANOVA was performed. Values are expressed as the mean ± S.E.M.

## Data Availability

Data are contained within the article.

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
