# Peer review of "Intestine Health and Barrier Function in Fattening Rabbits Fed Bovine Colostrum"

_vetsci, 2023, doi:10.3390/vetsci10110657_

Round 1

Reviewer 1 Report

Comments and Suggestions for Authors

In the manuscript titled "Intestine health and barrier function in fattening rabbits fed 2 bovine colostrum" by Lucia Aidos et al. The authors evaluated the beneficial effects of bovine colostrum in the post-weaning diet of farmed rabbits.

The complete text is well-designed, plain, and detailed in the methods and the presentation of the results. The choice of applied techniques is proper, and the antibodies used are suitable for the final purpose. Although some results may seem mixed, they also indicate the benefits of using bovine colostrum during the weaning of farm rabbits. The data obtained from this research are crucial to understanding the potential benefits of dietary factors on the development of the gastrointestinal function of farmed species poorly researched like rabbits, to ensure improvements to the livestock supply chain as a reduction of mortality, disease, and finally an increase in the growth rate in the fattening phase.

Author Response

The authors thank the reviewer for his comments. 

Reviewer 2 Report

Comments and Suggestions for Authors

The manuscript focuses on the effects of bovine colostrum as a supplement on the health of weaning animals, specifically in rabbits. Data on intestine histology and immunofluorescence were presented to demonstrate the effects on the intestinal barrier in terms of morphology, proliferation level, tight-junction expression, and enteric nervous system inflammation status. With high level of BC may lead to high intestinal permeability and possible inflammation, the authors concluded that an intermediate level of BC may have a positive tropic effect in weaning rabbits.

Major Comments:

1.       Please add descriptive titles for all result sections.

2.       Please explain the rationale for choosing 2.5% and 5% BC as supplements.

Minor comments:

3.       Line 15, “and non-specific enteritis and…” please edit this sentence.

4.       Figure 1: please consider using more distinctive representations for three groups.

5.       Figure 3: please consider using a larger font for the scale bar, currently, it is not readable. Same for other histological figures.

6.       Starting from the discussion section, all citations were in bold. Please make sure the format is consistent.

7.       Line 426-429, please edit this sentence. 

Comments on the Quality of English Language

Minor English edits are needed. 

Author Response

The authors thank the reviewer for the comments and suggestions. Please find attached our detailed answers.

Reviewer 3 Report

Comments and Suggestions for Authors

The work focuses on effects of to evaluate the effects of the bovine colostrum supplementation on the intestinal barrier such as intestine morphology and proliferation level, tight-junctions expression and enteric nervous system inflammation status in fattening rabbits. In general, the manuscript is well-written. However, some of the points listed below must be evaluated and corrected. Please consider all of the suggestions listed below.

1.     1. More information is needed for the conclusion and recommendation. Line 37: How did you conclude that including 2.5% BC in your diet could be a reasonable compromise between intestinal morphology and permeability? How about 5%? Since you have two levels of BC, they must be considered.

2.     2. The introductory section comprised 61 citations and should be shortened to concentrate only on why such parameters as intestinal barrier and enteric nervous system inflammation were examined and why the authors used current levels of 2.5% and 5% bovine colostrum in the trial.

3.     What are the underlying mechanisms that account for variations in the intestine's morphology and proliferation, tight junction expression, and enteric nervous system inflammatory status of rabbits fed bovine colostrum?  The discussion part has to be specific. You did not explain the underlying causes of these changes.    

Comments on the Quality of English Language

The work focuses on effects of to evaluate the effects of the bovine colostrum supplementation on the intestinal barrier such as intestine morphology and proliferation level, tight-junctions expression and enteric nervous system inflammation status in fattening rabbits. In general, the manuscript is well-written. However, some of the points listed below must be evaluated and corrected. Please consider all of the suggestions listed below.

1.     1. More information is needed for the conclusion and recommendation. Line 37: How did you conclude that including 2.5% BC in your diet could be a reasonable compromise between intestinal morphology and permeability? How about 5%? Since you have two levels of BC, they must be considered.

2.     2. The introductory section comprised 61 citations and should be shortened to concentrate only on why such parameters as intestinal barrier and enteric nervous system inflammation were examined and why the authors used current levels of 2.5% and 5% bovine colostrum in the trial.

3.     What are the underlying mechanisms that account for variations in the intestine's morphology and proliferation, tight junction expression, and enteric nervous system inflammatory status of rabbits fed bovine colostrum?  The discussion part has to be specific. You did not explain the underlying causes of these changes.    

Author Response

(The authors gave the same response as above.)
